

# Natural language understanding of map navigation queries in Roman Urdu by joint entity and intent determination

Javeria Hassan[1], Muhammad Ali Tahir[1] and Adnan Ali[2]

[1] National University of Sciences and Technology (NUST), Islamabad, Pakistan
[2] University of Science and Technology of China, Hefei, Anhui, China

## ABSTRACT

Navigation based task-oriented dialogue systems provide users with a natural way of communicating with maps and navigation software. Natural language understanding (NLU) is the first step for a task-oriented dialogue system. It extracts the important entities (slot tagging) from the user's utterance and determines the user's objective (intent determination). Word embeddings are the distributed representations of the input sentence, and encompass the sentence's semantic and syntactic representations. We created the word embeddings using different methods like FastText, ELMO, BERT and XLNET; and studied their effect on the natural language understanding output. Experiments are performed on the Roman Urdu navigation utterances dataset. The results show that for the intent determination task XLNET based word embeddings outperform other methods; while for the task of slot tagging FastText and XLNET based word embeddings have much better accuracy in comparison to other approaches.

## INTRODUCTION

A navigation dialogue/query system (*Zheng, Liu & Hansen, 2017*) is a salient use case in the task oriented dialogue systems domain. Its input is a navigational query which is written/spoken when a user is driving or walking. The input mode can be text as well as speech. This navigational query might include a particular point of interest (POI) as destination and the user's intent regarding that POI like directions to the POI or its distance. To retrieve the POI from the input navigation query/utterance and determining the user's intent is the task of the natural language understanding (NLU) module (*Yao et al., 2013*; *Mesnil et al., 2015*). This module is a part of the task oriented dialogue system. The natural language understanding module of any dialogue system includes three tasks which are domain detection, slot tagging and intent determination. The output of this module includes the intent and slots of the input utterance; these are called dialogue states. Dialogue states are used to query our knowledge base or an external database and return some sort of output. In case of a navigation oriented dialogue system, the slots may contain a destination point and the user's intent could be finding directions to that destination.

Figure 1 shows the pipeline framework of a navigation oriented dialogue system. It can be seen that NLU is a central step in the task oriented dialogue system; therefore, the

Corresponding author
Javeria Hassan,
jhassan.mscs17seecs@seecs.edu.pk

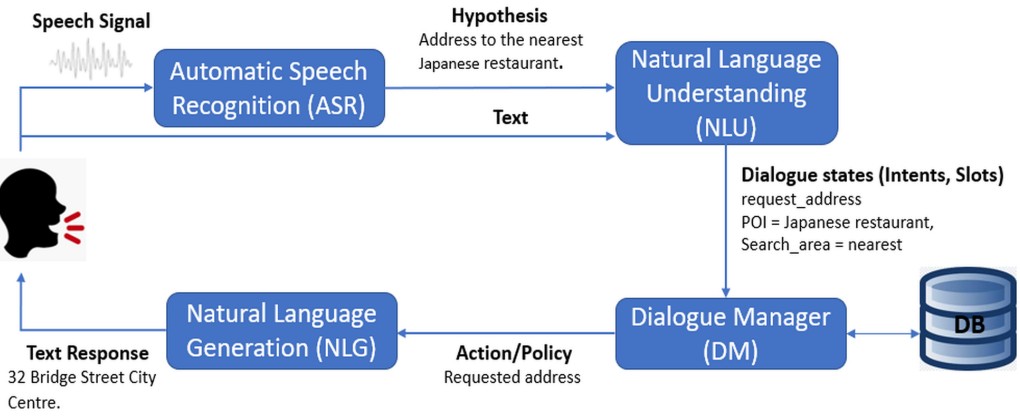

**Figure 1   Pipeline framework of navigation oriented dialogue system.**

accuracy of natural language understanding greatly affects the output of the whole dialogue system.

Figure 2 shows the natural language understanding process for an input utterance. Intent determination task is similar to the multi-class classification task. The slot tagging task is generally more challenging than intent determination. It is similar to sequence classification where the classifier's job is to determine the semantic tags for the sub-sequences in the utterance. The tasks of slot tagging and determining the intent of an input utterance are somewhat different from each other. During the pre-deep learning era they were modeled using separate approaches; like support vector machines for intent determination and conditional random fields for slot tagging. Now, using deep learning, it is possible to determine the solution of both tasks jointly using a single model. *Hakkani-Tür et al. (2016)* proposed joint model for intent determination, slot tagging and domain detection using the RNN-LSTM architecture. The input of this model are the user utterances while the output includes the domain intent and slots. The main principle of joint modeling is similar to that of the sequence-to-sequence modeling (*Sutskever, Vinyals & Le, 2014*) or neural conversational model (*Vinyals & Le, 2015*); because the last hidden layer of the neural network contains the semantic information of the whole input, giving us intent and domain information. *Liu & Lane (2016)* proposed an LSTM based encoder–decoder. The model is attention based with a bidirectional encoder and a unidirectional decoder. This model jointly models the intent determination task and slot tagging task. Similarly, *Goo et al. (2018)* also proposed an attention-based encoder–decoder architecture but they included a slot gate. The purpose of the slot gate is to make use of the intent based context vector, while determining the slot labels for an utterance. Slot gate based model outperformed simple attention-based encoder decoder architecture for joint modeling of the slot tagging and intent determination. *Hardalov, Koychev & Nakov (2020)* proposed joint intent detection and slot tagging, which was built upon a pre-trained BERT language model. It first determines the intent distribution vector by adding an additional pooling layer to get a hidden representation of the entire input utterance, then it obtains the predictions for each token in utterance using BERT language model. Both of these are

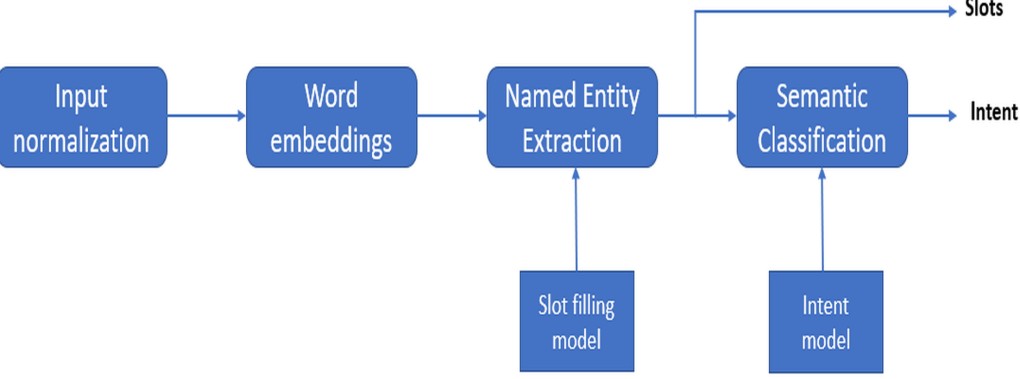

**Figure 2** **Block diagram of natural language understanding process.**

used as input to predict the slots of an input utterance. The above mentioned methods utilize the intent to determine the slots. Conversely, it can also be beneficial if we use the determined slots for intent determination. *Peng et al. (2020)* proposed an interactive two-pass decoding network. It is the joint slot tagging and intent determination model. It uses first pass decoder to determine the explicit representation for the first task; and then uses this representation as an input for the second pass decoder to determine the results of the second task. This model takes full advantage of both of the determined intents and slots, so that it can achieve bidirectional conversion between these two tasks. Joint modeling provides a major advantage in comparison to the separate models for slot tagging and intent determination. It provides higher accuracy with a smaller amount of labeled data. Therefore, in our work, we have implemented a joint model as proposed by *Liu & Lane (2016)*.

When it comes to natural language understanding of the navigational dialogues, there is not a lot of work done in Urdu language. This is especially true for Roman Urdu and deep learning techniques. Roman Urdu here refers to writing Urdu language using the transliteration in Roman (English) alphabet; as opposed to the standard way of writing Urdu in Arabic script (with some extra letters). Examples of Roman Urdu sentences can be seen in Fig. 3. Roman Urdu has been popularized in the last 2 decades due to increased use of Urdu writing on the internet and mobile phones using the standard English keyboard. Though there are examples of deep learning applied to Roman Urdu text, those are in NLP domains other than natural language understanding, like sentiment analysis (*Ghulam et al., 2018*; *Shakeel & Karim, 2020*) and Roman Urdu to Urdu transliteration (*Alam & ul Hussain, 2017*). In this research, we are going to work on natural language understanding of a Roman Urdu navigational dialogue dataset. The input sequence to the natural language understanding model is converted into word embeddings as can be seen in Fig. 2. Word embeddings are the distributed vector representations of words in a document, which capture the semantic and syntactic meanings of these words. There are mainly two types; context-independent and context-dependent word embedding methods. The Word2Vec (*Mikolov et al., 2013*) model was the first neural network based model, which maps the

| Roman Urdu (Sentences in dataset) | Urdu Equivalent | English Translation |
|---|---|---|
| Meine Saddar Rawalpindi jana hai. | میں نے صدر راولپنڈی جانا ہے۔ | I want to go to Saddar Rawalpindi. |
| Qareeb tareen petrol pump kahan hai ? | قریب ترین پٹرول پمپ کہاں ہے ؟ | Where is the nearest petrol pump ? |
| Street 5 kahan hai sector F mein ? | سٹریٹ 5 کہاں ہے سیکٹر ایف میں ؟ | Where is street 5 in sector F ? |
| Mein Lahore ki taraf jana chahta hoon. | میں لاہور کی طرف جانا چاہتا ہوں۔ | I want to go towards Lahore. |

**Figure 3** Example sentences from Roman Urdu navigation dataset, along with their regular Urdu equivalent and English translation.

words to their distributed representation while capturing the syntactic and semantic meaning of the words. An extension to the Word2Vec model FastText was proposed by *Bojanowski et al. (2017)*; which is better at predicting and recognizing out-of-vocabulary words in comparison to the Word2Vec model. A major drawback of the FastText and Word2Vec models is that both are context-independent. In context-independent methods, the order of the words does not effect the resulting word embedding. To take advantage of the context information, deep learning based models have been introduced. Elmo was proposed by *Peters et al. (2018)*; the model architecture includes the bidirectional LSTM and CNN. It has the ability to capture the word meanings with changing context. Google introduced BERT (*Devlin et al., 2019*), which produces embeddings in a similar manner to those of ELMO. BERT is based on a deep learning based architecture known as transformer (*Vaswani et al., 2017*). Transformer is an encoder–decoder based model which includes multi-head attention in both encoder and decoder layers. XLNET (*Yang et al., 2019*) is an extension to BERT. It is also based on the transformer, but it introduces the permutation language modeling, which predicts its tokens in random order rather than left to right as in BERT. *Ghannay, Neuraz & Rosset (2020)* have studied the effect of different word embedding approaches on the natural language understanding output. They have compared both context independent and context dependents approaches and then used these embeddings as an input to the Bidirectional LSTM for natural language understanding. The datasets that they have used include both large and small corpora. Results have clearly shown that the embeddings based on the larger datasets had better accuracy compared to smaller corpora. In this paper we are going to compare different word embedding methods; these are FastText, ELMO, BERT and XLNET. We are also investigating transformer based approaches. The semantic representations produced by these approaches will be evaluated, and it will be observed how these semantic representations effect the accuracy of the joint intent detection and slot tagging model. Main objectives of our paper are: (1) Comparison of word embeddings created by different methods for joint slot tagging and intent determination model, and their effect on the

F1-score, and; (2) natural language understanding of navigational dialogues in Roman Urdu using a joint slot tagging and intent determination model.

The rest of the paper is organized as follows. 'Methods' explains different concepts such as natural language understanding and word embeddings, as well as various machine learning models used for their determination. 'Results' introduces the data set, the experimental methodology and results obtained. Finally, 'Discussion' provides a conclusion and future prospects.

## METHODS

### Natural language understanding model

A joint slot tagging and intent determination model will be used. The reasoning behind using this joint model is that it provides better accuracy with a smaller number of labeled sentences in training data. We are going to use an LSTM based encoder decoder model with attention mechanism and the aligned inputs (*Liu & Lane, 2016*).

**Attention based Encoder-Decoder model** Encoder–decoder architecture including attention and aligned inputs as shown in Fig. 4 provides higher accuracy for slot tagging and intent determination tasks (*Liu & Lane, 2016*). The model is an LSTM based encoder–decoder. LSTM is used as the basic recurrent network unit because it models long-term dependencies better than the simple RNN. The model includes the BLSTM (*Graves & Schmidhuber, 2005*) encoder; it reads the input word sequence $x = (x_1, x_2, x_3, \ldots, x_T)$ in the forward and backward directions. It generates the hidden state $h_f$ while reading the input sequence in the forward direction and the hidden state $h_b$ while reading the input sequence in the backward direction (i.e., opposite to that of the original order of input word sequence), for the time step $i$. The hidden state of the encoder $h_i$ at time step $i$ is computed by the concatenation of the $h_f$ and $h_b$ hidden states,

$$h_i = [h_f, h_b]. \tag{1}$$

The model includes two decoders; one for intent determination task and one for slot tagging task. Slot tagging decoder output includes the labels $y = (y_1, y_2, y_3, \ldots, y_T)$ for each of the words in the input sequence. The intent determination decoder produces a single output which is the intent of whole input word sequence. The slot tagging decoder is LSTM based. The decoder state $s_i$ at time step $i$ is a function of the previous output $y_{i-1}$, previous decoder state $i-1$, aligned encoder hidden state $h_i$ and context vector $c_i$ which is attention (*Bahdanau, Cho & Bengio, 2015*).

$$s_i = f(y_{i-1}, s_{i-1}, h_i, c_i) \tag{2}$$

$$c_i = \sum_{j-1}^{T} \alpha_{i,j} h_j \tag{3}$$

$$\alpha_{i,j} = \frac{\exp g(s_{i-1}, h_j)}{\sum_k^T \exp g(s_{i-1}, h_k)}. \tag{4}$$

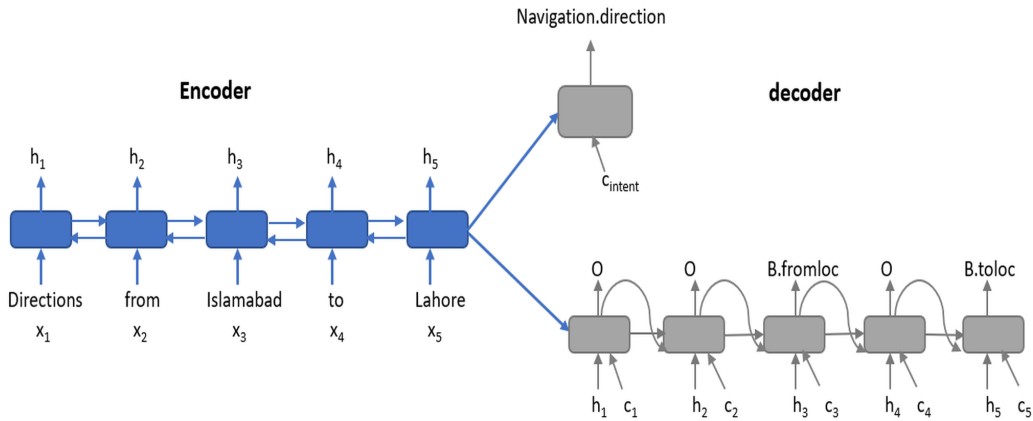

**Figure 4  Attention based encoder decoder for the joint modeling of intent determination and slot filling.**

The context vector $c_i$ is the weighted sum of the hidden encoder states. To find the weights to assign to these hidden states, a feed forward network $g$ is trained. This network uses the previous state of decoder and all the encoder hidden states to calculate the output encoder state. The intent determination decoder state is determined by the $c_{intent}$ context vector and $s_0$ initial decoder hidden state. $s_0$ carries information of the entire input word sequence.

## Word embeddings

Word embedding models map the words to distributed representations which capture the semantic and contextual meaning of the words. For this work, word embeddings have been created using four different methods. These are described below.

**FastText** FastText is an extension of Word2Vec model, introduced by the Facebook AI (*Bojanowski et al., 2017*). FastText has a lot of advantages over GloVe and Word2Vec. First of all it has the ability to capture the semantics of shorter words better. It is also comparatively better at recognizing and predicting out-of-vocabulary words. FastText model represents the words in the form of n-grams of their characters, and then it trains the skip-gram model to learn the embeddings. The values produced for each n-gram representation are summed up to form one vector during each training step. Facebook has provided pretrained models for 157 languages including Urdu.

**ELMO** ELMO (*Peters et al., 2018*) stands for Embeddings from Language Models. It has the ability to capture deep contextualized meanings of the words in its embeddings. Its architecture includes multiple character based CNN's, and on top it has bidirectional LSTM to model the whole sentence. Thus, the main task of the bidirectional LSTM is to create the contextual word embeddings. ELMO creates character based representations—this naturally makes it more robust to the out-of-vocabulary words. ELMO creates the word vector representations at run-time.

**BERT** BERT (*Devlin et al., 2019*) architecture includes multiple layers of the bidirectional transformer encoder. The BERT model provides highly powerful context based word representations. Transformer (*Vaswani et al., 2017*) is an encoder–decoder based model

which includes multi-head attention. It completely relies on self-attention for computing its input and output representations, without relying on the aligned RNNs. The BERT's input representation is basically a concatenation of positional embedding, segment embedding and word piece embedding. For fine tuning there are special tokens - [CLS] is inserted as the first token, [SEP] is a special token which is added at the end as the last token, and [MASK] is only used for pre-training and for masked words.

**XLNET** XLNET (*Yang et al., 2019*) is a generalized autoregressive pre-training model. XLNET outperforms BERT in many natural language processing tasks. Autoregressive model is a feed-forward network, which determines the future words from a given set of words by either using the forward context or backward but not both. It is called a generalized autoregressive model because it uses permutation language modeling to capture the bidirectional context. Permutation language modeling predicts its tokens in random order, rather than left to right as in BERT.

## Dataset

We have created and used a navigational utterances dataset in Roman Urdu. Example sentences from this dataset can be seen in Fig. 3. To our knowledge, such a dataset based upon navigational dialogue in Roman Urdu has not been collected before. We have similar such datasets in other languages especially in English like multi-turn, multi-domain dialogue dataset (*Eric & Manning, 2017*), Atis (*Bungeroth et al., 2008*), and CU move (*Hansen et al., 2005*). Roman Urdu is more commonly used among Pakistani people for short text messaging (SMS) and on social media platforms, in comparison to either English Language or original Urdu script. Furthermore, Pakistani street addresses available from Google API are also in Roman Urdu. Therefore, if we want to create an online text based dialogue agent, people will find it much easier to communicate with it in Roman Urdu rather than communicating in English language or original Urdu script. Our system can be interfaced with a speech recognition based front-end to create navigational dialogue system to help the drivers on the road. There are a large number of drivers with limited English language skills; who would be more comfortable in using a navigational dialogue system in Urdu language rather than English. The dataset was collected from Pakistani university students, with ages between 18 and 22 years. This group of subjects was chosen because these students are tech savvy and frequent users of maps and navigation software. Their opinion (training set examples) would give a good estimate of an average user of text or voice based maps applications. These questions were related to the issues that they face while driving or generally looking for locations in an unfamiliar area. After the collection of dataset, the next step is the pre-processing. The main issue with Roman Urdu is that everyone has their own spelling style; this would cause problems for creating word embeddings of the dataset. To mitigate this problem lexical normalization is applied to the dataset. The next step is dataset annotation i.e., assigning the intent to each utterance and slot labels to each of the words in an utterance, as can be seen in Table 1. The slot labels are assigned based on the IOB (Inside Outside and Beginning) format, which is a common NER (named entity recognition) format. An example of an annotated utterance is given below. 21 distinct intent labels and 29 distinct slot labels have been assigned.

**Table 1  Example of the navigational query in IOB format.**

| Sentence | Context |
|---|---|
| Directions | B-directions |
| from | O |
| Lahore | B-fromloc |
| to | O |
| Islamabad | B-toloc |

# RESULTS

We have trained and tested our models on the Roman Urdu navigational dialogues dataset. Details of the experimental setup are given below.

## Training details

**Attention based encoder decoder** In this model LSTM is used as the basic RNN unit. The number of units in the LSTM cell is set to 200. The number of layers for LSTM is set to 1. Dropout rate is 0.5 and learning rate is 0.001. Maximum norm is set to 5. The model has been trained for 50 and 100 epochs.

**FastText** FastText Urdu predtrained model has been used for creating word embeddings. The size of the word embeddings is 300.

**ELMO** We used a pretrained ELMO model available. It has 2 LSTM layers with 1024 hidden states for each layer and character based word representation vector of size 512.

**BERT** *bert-base-multilingual-cased* model has been used. It is trained on 104 languages including Urdu. This model is 12 layered, with 768 hidden states and 12 heads.

**XLNET** *xlnet-base-cased* model is used. This model is 12 layered, and has 768 hidden states and 12 heads.

## Results

### Evaluation metrics

The evaluation metric used to evaluate the model is the F1 score, which is the harmonic mean between the precision and recall. If we talk in terms of our dataset; the intent determination model assigns each sentence an intent (navigate-directions, navigate-time, navigate-search-directions). Similarly, precision determines the overall number of correctly determined intents out of all the predicted intents. The data is unevenly distributed; therefore we cannot only rely on the number of correctly predicted classes. If we only rely on precision our results will only focus on the commonly present intents in the dataset (like navigate-loc.search), and not focus on the other important intents (like navigate-time). This is why recall measure is needed, as it determines how many different intents are correctly classified. If precision is high, the recall may be low. An ideal model would have both of these metrics balanced. Therefore, F1 score has been used as it balances the trade-off between both precision and recall. Furthermore, it works really well when it comes to the unevenly distributed multi-class dataset like ours.

Word embeddings have been created on our dataset using four of the above mentioned models. Each of these embeddings are then used as an input to the attention based

**Table 2  Validation and evaluation F1 scores (%) after 100 epochs of intent determination.**

| Model | Validation | Evaluation |
|---|---|---|
| FastText | 80.00 | 76.00 |
| ELMO | 82.00 | 80.00 |
| BERT | 76.00 | 71.00 |
| XLNET | 84.00 | 84.00 |

**Table 3  Validation and evaluation F1 scores (%) after 100 epochs of slot tagging.**

| Model | Validation | Evaluation |
|---|---|---|
| FastText | 76.11 | 82.24 |
| ELMO | 82.11 | 79.74 |
| BERT | 78.33 | 79.57 |
| XLNET | 79.59 | 81.81 |

encoder–decoder model. The model determines both the intent and slots of an input dataset. Given below is the comparison of the F1 scores of the models for each of the input word embeddings for both intent determination and slot tagging.

### Intent determination

In Table 2, F1 scores of the intent determination models are given after 100 epochs. From evaluation results it can be seen that F1 score for the word-embedding based on XLNET model has outperformed the other methods for intent determination; with the F1 score for evaluation being 84.00.

### Slot tagging

In Table 3, F1 score of the model for slot tagging is given after 100 epochs. Looking at the evaluation F1 scores for all the models, it can be seen that both XLNET and FastText based word embeddings have higher F1 scores in comparison to the other models; with evaluation F1 scores being 82.11 and 82.24 respectively.

## DISCUSSION

Our dataset was human labeled training data. One problem with this type of human labeled datasets is that they are prone to errors. Another issue with Roman Urdu is that there are no standardized spellings; different writers may use different spellings for the same word. Even though we have normalized each word to one spelling; still there are a few words having more than one spellings. This leads to poor generalization capability for word embedding models.

### Intent determination

For the task of intent determination, there are 21 distinct classes. The model assigns each utterance an intent from those distinct classes. If we look at the F1 score (%) plot in Fig. 5A it can be clearly seen that the word embeddings which were created using the XLNET model has the highest F1 (%) of 84.00. Figure 6A contains the confusion matrix, based of the

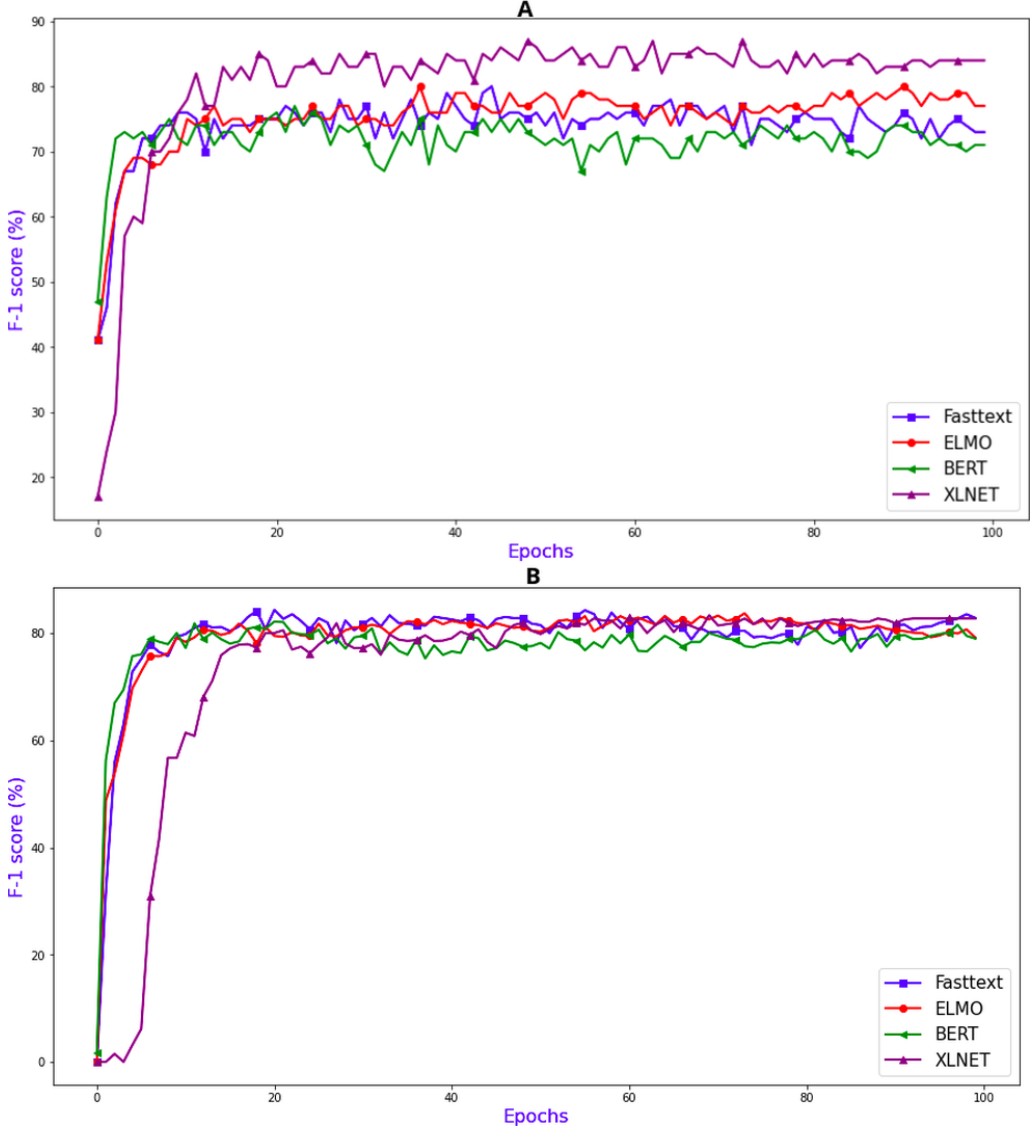

**Figure 5** Evaluation F1 score (%) graphs (A) for intent determination and (B) for slot tagging.

intent classes predicted using the word embedding created using XLNET. The diagonal of the confusion matrix shows the number of classes predicted correctly. XLNET based word embeddings have provided much better results for the intent determinations task, because it is much better at capturing the contextual information present in an utterance. XLNET is a bidirectional transformer. If we talk about the F1 (%) plot in Fig. 5A, it can be seen that in comparison to BERT (which is also transformer based), XLNET performs much better for this task. XLNET uses the permutation language modeling, and predicts all the token in random order. This helps it to better understand the bidirectional relationships among the words in comparison to BERT which only predicts 15% of the masked tokens.

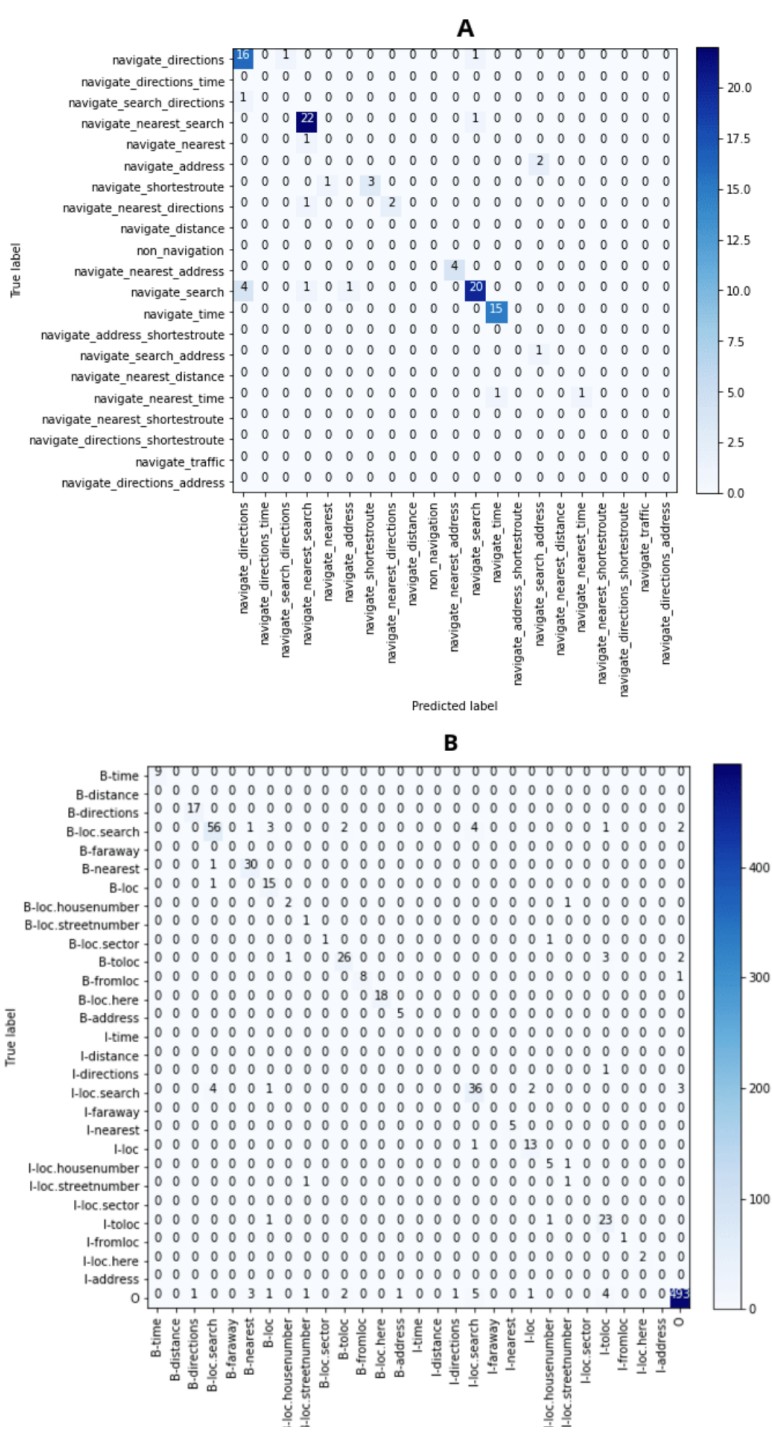

**Figure 6** Confusion matrix (A) for intent determination (XLNET) and (B) for slot tagging (FastText).

## Slot tagging

For the task of slot tagging, there were 29 distinct classes. Our model assigned a slot tag
to each of the word in an utterance. If look at the F1 scores (%) plot in Fig. 5B it can

be clearly seen that the word embeddings created using the FastText model and XLNET performed really well with the F1 score of 82.24 and 81.81. Figure 6B contains the confusion matrix, based on the slot tags using the word embedding created using FastText. F1 score is dependent upon the number of classes correctly identified. FastText is better at recognizing the out-of-vocabulary words. FastText model represents the words in the form of n-grams of their characters. The n-grams of characters are learned such that the sum of these character representations is equivalent to the word representation. So the words which are missing some letters or out-of-vocabulary words can learned because they are represented using the sum of the representations of the n-gram characters contained in those words. XLNET has provided good accuracy. However it could not perform as good as it did for the intent determination task, because the dataset is too small for the task of slot tagging. Transformer based models are generally much better with larger datasets. If we look at the F1 plot for slot tagging, ELMO based word embedding also provided much better accuracy for earlier epochs and also has the highest validation F1 score 82.11. ELMO creates word distributed representations using deep learning models. It is character-based like FastText, therefore it is great at context based modeling and for modeling the out-of-vocabulary words.

*Ghannay, Neuraz & Rosset (2020)* also studied the effect of word embeddings on NLU. They have compared GloVe, FastText, and ELMO. For the slot tagging and intent determination tasks they have used the bidirectional LSTM encoder–decoder model. Their experimental results have shown that the word embeddings created using the larger out-of-domain datasets yield better results in comparison to smaller datasets. Their results have shown that even for the larger out-of-domain datasets the embeddings created using ELMO provided the highest score. They did not compare the transformer based models. In our research work, we have also explored transformer based word embedding approaches. Furthermore, we have also used the attention mechanism in the slot tagging and intent determination tasks.

## CONCLUSIONS

In this paper we have used a joint slot tagging and intent determination model for for determining the slots and intent of navigational queries in Roman Urdu. We have used different approaches for creating word embeddings of our dataset. We wanted to determine how the word embeddings created using different approaches will effect the results of slot tagging and intent determination models. Word embeddings were created using the both context independent and dependent methods. The experimental results have shown that for the intent determination task the word embeddings created using the XLNET provided much better F1 score. XLNET is a transformer based model and more effective at capturing the relationships and dependencies among words in an utterance as compared to other approaches. For the task of slot tagging, word embeddings created using XLNET and FastText provided much better results. FastText has the ability to cater to rare/out-of-vocabulary words much better because it creates representations for words not present at training time. ELMO also provided the highest validation score for the task

of slot tagging. ELMO is based on Bidirectional LSTM and CNN, and provides much better context based representation for the resulting embeddings. Future work in this direction is suggested to focus on gathering larger data sets, as the performance of some methods like BERT could be more pronounced on large datasets. Also, having more demographic variation in the data gathering subjects could lead to newer insights.

### Funding
The authors received no funding for this work.

### Competing Interests
The authors declare there are no competing interests.

### Author Contributions
- Javeria Hassan conceived and designed the experiments, performed the experiments, analyzed the data, prepared figures and/or tables, and approved the final draft.
- Muhammad Ali Tahir analyzed the data, prepared figures and/or tables, authored or reviewed drafts of the paper, and approved the final draft.
- Adnan Ali analyzed the data, authored or reviewed drafts of the paper, and approved the final draft.

### Data Availability
The code is available at GitHub:
https://github.com/sz128/slot_filling_and_intent_detection_of_SLU.git.
The Python code files are available in the Supplementary File.

### Supplemental Information
Supplemental information for this article can be found online at http://dx.doi.org/10.7717/peerj-cs.615#supplemental-information.

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
