# Peer review of "Natural language understanding of map navigation queries in Roman Urdu by joint entity and intent determination"

_PeerJ Computer Science, doi:10.7717/peerj-cs.615_

## Round 0.1 · original submission · Major Revisions

Revise the manuscript following the suggestions and comments of the reviewers.

Reviewer 1 ·

Basic reporting

1. Please double-check the grammar and language to make sure the delivery is clear, such as the sentence at line 43 – 45 needs to rearrange.
2. Please double-check the grammar, lots of Singular nouns are missing their articles.
3. Check the line number; the line numbers are break after line 122.

Experimental design

a. The dataset introduction should be in the method part, which would give the audience a clearer idea of how the experiment is designed.
b. Please give justifications for the selection of the datasets. Is it the only available one, or is there a reason that this specific dataset is selected? Is it based on the property of the dataset or based on the method?
c. If you are using F-score as an estimation for the performance, please explain how they work and the possible merit and drawback of using F-score as an estimation method other than other methods?
d. Please give a more detailed description and comparison of results from different methods and from different epochs. Is this outcome your prediction, and what are the possible causes for the difference between results? It is not presented clearly in this paper how the different methods would impact the F1 Score.

Validity of the findings

This data and results in this paper are robust and sound. However, the justification of why this dataset is brought into use needs to be clearer.

Reviewer 2 ·

Basic reporting

The idea of the paper is sound and need be investigated more however few suggestions are there .
1 abstract needs to be rewritten with respect to findings
2 add latest literature in this area
3 evaluation of proposed method is not clear therefore authors need to provide confusion matrix to understand the recognition rate of the system.
Therefore I recommend to accept the paper after incorporating above comments

Experimental design

Experiments need to be explained more

Validity of the findings

Need to justify the results with latest literature

Additional comments

Authors are suggested to proof read from native English speaker

---

## Round 0.2 · Minor Revisions

Revise the manuscript following the suggestions and comments of the reviewer.

Reviewer 1 ·

Basic reporting

It can be seen that after the modification, the language level and quality of this paper have been improved.But in our opinion, this paper still has some small shortcomings that need to be revised.
1. The "Abstract" part of the paper needs to be condensed.Too much technical background.
2. This paper has introduced many references in the "Introduction" section, but I think what should be introduced is the relationship between these literatures and your research, rather than simply listing the work done by these literatures.
3. The quality of Figure 5 needs to be improved.It is suggested that the meaning of each curve be intuitively expressed in the picture.
I hope the author can seriously modify the article and polish the language in the article. After the quality is further improved, it can be considered for publication.

Experimental design

no comment

Validity of the findings

no comment

Additional comments

no comment

---

## Round 0.3 · accepted · Accept

The paper is accepted based on the recommendation of the reviewer.

Reviewer 1 ·

Basic reporting

The authors have pay attention to all the advice. Suggest to publish.

Experimental design

The paper is original and fitful for the journal. The author has respond to all the advices.

Validity of the findings

This paper is validity and provide novel contributions to the field. The authors have made all the adjustment required.

Additional comments

The author has respond to all the advices. Suggest to publish.